**Data Availability Statement:** The dataset and a copy of the survey are available in Arch, Northwestern University's Institutional Repository (https://doi.org/10.21985/n2-00h0-a153).

# Motivators and barriers to research participation for individuals with cerebral palsy and their families

Kristina M. Zvolanek[1,2‡], Vatsala Goyal[1,2‡], Alexandra Hruby[1,2], Carson Ingo[1,3], Theresa Sukal-Moulton[1,4]*

1 Department of Physical Therapy & Human Movement Sciences, Feinberg School of Medicine, Northwestern University, Chicago, IL, United States of America, 2 Department of Biomedical Engineering, Northwestern University, Evanston, IL, United States of America, 3 Department of Neurology, Feinberg School of Medicine, Northwestern University, Chicago, IL, United States of America, 4 Department of Pediatrics, Feinberg School of Medicine, Northwestern University, Chicago, IL, United States of America

‡ KMZ and VG are share first authorship on this work.

* theresa-moulton@northwestern.edu

## Abstract

### Objective(s)

Our objective was to investigate the motivators and barriers associated with the individual or family decision to participate in cerebral palsy research. Based on this information, we offer suggestions to increase the likelihood of participation in future CP studies.

### Methods

A digital survey was administered to stakeholders affected by cerebral palsy across the US. Our analysis focused on variables related to personal interests, travel, and study-specific elements. Statistical tests investigated the effects of responder type, cerebral palsy type, and Gross Motor Function Classification System level on travel and study-specific element variables. Recommendations were informed by responses reflecting the majority of respondents.

### Results

Based on 233 responses, we found that respondents highly valued research participation (on average 88.2/100) and compensation (on average 62.3/100). Motivators included the potential for direct benefit (62.2%) and helping others (53.4%). The primary barriers to participation were schedule limitations (48.9%) and travel logistics (32.6%). Schedule limitations were especially pertinent to caregivers, while individuals with more severe cerebral palsy diagnoses reported the necessity of additional items to comfortably travel.

### Conclusions

Overall, we encourage the involvement of stakeholders affected by cerebral palsy in the research process. Researchers should consider offering flexible study times,

**Funding:** This work was supported by National Institutes of Health (https://urldefense.com/v3/__ https://grants.nih.gov/funding/index.htm__;!! Dq0X2DkFhyF93HkjWTBQKhk!HIG4deujDRJ 5GWkKeob1oKU1NMPw4GhqH5WRaJJj5s6F rZ77ZhhVhddionuJYRMnCDaxd8lwCqzL$) awards R01NS058667 supporting TSM and CI, R03HD094615 to CI, a predoctoral training fellowship T32EB025766 to KMZ, a predoctoral training fellowship T32HD007418 to VG, and a predoctoral training fellowship T32EB009406 to AH.

**Competing interests:** The authors have declared that no competing interests exist.

**Abbreviations:** CP, cerebral palsy; GMFCS, Gross Motor Function Classification System; REDcap, Research Electronic Data Capture.

accommodating locations, and compensation for time and travel expenses. We recommend a minimum compensation of $15/hour and a maximum time commitment of 4 hours/day to respect participants' time and increase likelihood of research participation. Future studies should track how attitudes toward research change with time and experience.

## Introduction

Cerebral palsy (CP) is a broad pediatric-onset diagnosis caused by a non-progressive injury to the developing brain [1]. The etiology of CP is also extremely heterogeneous, often resulting from brain injuries that occur during the early developmental period [2–5]. Although CP is considered a pediatric-onset disorder, the associated physical and behavioral presentations are present across the lifespan and may fluctuate in severity over time. As there is currently no cure for CP, research efforts are critical for advancing our understanding of the pathophysiology and most efficacious treatments. However, the diversity of this population poses a significant recruitment challenge to researchers [6]. Limited funding for CP research also puts an added burden on researchers to be efficient with study-specific elements and recruitment, especially in the United States [7]. In an attempt to address this, previous studies have surveyed individuals with CP to identify priority research areas [8–10]. Research registries have also been established to facilitate collaboration among US institutions and to improve communication between researchers and individuals with CP [11, 12]. Despite these efforts, the success of CP research is dependent on the desire of individuals to participate and their ability to reasonably access the study within the limitations of their environment.

A previous study investigated the barriers to intervention-based CP research and recommended the involvement of patient populations and their families in the study development pipeline to improve recruitment [6], an approach towards community-based participatory research. However, the study did not consider facilitators to research and only evaluated a small subset of the population interested in home-based training programs. Therefore, our objective was to sample a larger and more heterogeneous cohort of stakeholders and investigate the motivators and barriers associated with the decision to participate in CP research. We aimed to inform researchers of specific stakeholder perspectives by understanding whether factors such as Gross Motor Function Classification System (GMFCS) level, CP type, or age contribute to the decision to participate. Based on a nationwide survey, we provide recommendations for investigators to increase likelihood of recruitment and participation in future CP research.

## Methods

### I. Recruitment

The survey, including informed consent, was created and administered using the Research Electronic Data Capture (REDCap) platform. It was approved by the Northwestern University Institutional Review Board and remained open between May 6th and July 7th, 2020. Respondents were eligible to voluntarily participate if they resided in the United States and were either 1) caregivers of minors (under 18 years of age) with a diagnosis of CP or 2) adults with a diagnosis of CP. The survey link was shared via several platforms, including the Cerebral Palsy Research Registry [11], ResearchMatch.org, department social media accounts, and emails to previous research participants.

## II. Experimental protocol: Qualitative survey

The digital open survey was designed to collect data about motivators and barriers of participation in CP research. The study objectives and survey questions were originally conceptualized from organic discussions among our research team. During survey development, we sought feedback from six caregivers of minors with CP and adults with CP to gather their opinions on the clarity of questions, the completeness of content, and the importance of the survey goals. The online survey [13] contained optional questions in six different categories: demographics, personal interests, travel needs & preferences, study-specific elements, past research experience, and impact of COVID-19. The present analysis focused on the first four categories to summarize attitudes towards general CP research. Details of the survey development are described in further detail by Joshi et al. [14].

## III. Survey categories

**Demographics.** We collected a number of variables to describe features of the respondents, listed in Table 1. In addition to CP type and GMFCS level, we asked about elements such as the time it takes for respondents to get to medical appointments and information about current and previous medical treatments common to study inclusion or exclusion criteria.

**Personal interests.** All subsequent variable names are italicized in text and described in Table 2. Personal interest factors were considered intrinsic to the respondent. Respondents were asked about their perception of *research importance*, how highly they *value research participation*, and *compensation importance*, all using a visual analog scale of 0–100 to easily quantify these subjective opinions. Open-ended questions in the survey requested comment on personal goals, motivators, and barriers for participation in CP research. To gauge specific research interests, respondents were asked what *study types* and *body functions* were of high interest to them.

**Travel needs & preferences.** Respondents reported whether additional arrangements for *childcare* would be required, whether *time off work* would be required, and what *additional travel needs* would be required when leaving the house. To understand travel preferences, respondents were asked about their typical mode of *transportation* to medical appointments. We also asked participants to identify their perceived maximum *travel time for indirect benefit study*, defined as a study that is seeking to understand more about CP, and maximum *travel time for direct benefit study*, defined as a study with the potential to offer a direct benefit to the participant. Participants were asked about their overall willingness to make an extended *overnight trip* for research participation. Finally, the importance of *travel reimbursement for local study* and *travel reimbursement for distant study* was evaluated, both on a scale from 0–100.

**Study-specific elements.** There were a number of variables related to explicit design of the study, which have the potential to be modified by the researcher. Respondents were asked about their most preferred study *locations* and their preferred *time of year* for study participation. Respondents' desired *compensation amount* was evaluated per hour of study participation. Respondents were also asked about the *maximum time commitment* that was reasonable for one day of participation, the *maximum study visits* they would be willing to commit, and whether they would consider participating in a *longitudinal* study.

## IV. Data and statistical analysis

IBM SPSS Statistics version 26 (IBM Corp., Armonk, NY, USA) was used to perform all analyses on the survey responses. Participants with missing data for a given question were

**Table 1. Participant demographics.**

| Characteristic | Respondent | | | | | |
|---|---|---|---|---|---|---|
| | All (n = 233) | | Adult with CP (n = 92) | | Parent of minor with CP* (n = 141) | |
| **Sex assigned at birth** | | | | | | |
| Male | 112 | 48.1% | 31 | 13.3% | 81 | 34.8% |
| Female | 120 | 51.5% | 61 | 26.2% | 59 | 25.3% |
| Not Reported | 1 | 0.43% | 0 | 0.00% | 1 | 0.43% |
| **Ethnicity** | | | | | | |
| Hispanic or Latino | 23 | 9.87% | 7 | 3.00% | 16 | 6.87% |
| Not Hispanic or Latino | 204 | 87.6% | 81 | 34.8% | 123 | 52.8% |
| Not Reported | 6 | 2.58% | 4 | 1.72% | 2 | 0.86% |
| **Race** | | | | | | |
| American Indian or Alaskan Native | 2 | 0.86% | 1 | 0.43% | 1 | 0.43% |
| Asian | 9 | 3.86% | 3 | 1.29% | 6 | 2.58% |
| Black or African American | 24 | 10.3% | 10 | 4.29% | 14 | 6.01% |
| Native Hawaiian or Other Pacific Islander | 1 | 0.43% | 0 | 0.00% | 1 | 0.43% |
| White | 175 | 75.6% | 68 | 29.2% | 107 | 45.9% |
| Two or More Races | 6 | 2.58% | 4 | 1.72% | 2 | 0.86% |
| Not Reported | 16 | 6.87% | 6 | 2.58% | 10 | 4.29% |
| **Gross Motor Function Classification System** | | | | | | |
| Level I | 60 | 25.8% | 15 | 6.43% | 45 | 19.3% |
| Level II | 65 | 27.9% | 33 | 14.2% | 32 | 13.7% |
| Level III | 33 | 14.2% | 23 | 9.87% | 10 | 4.29% |
| Level IV | 37 | 15.9% | 16 | 6.87% | 21 | 9.01% |
| Level V | 35 | 15.0% | 4 | 1.72% | 31 | 13.3% |
| Not Reported | 3 | 1.29% | 1 | 0.43% | 2 | 0.86% |
| **Cerebral Palsy motor topography** | | | | | | |
| Hemiplegia | 72 | 30.9% | 18 | 7.72% | 54 | 23.2% |
| Diplegia | 60 | 25.8% | 37 | 15.9% | 23 | 9.87% |
| Quadriplegia | 76 | 32.6% | 24 | 10.3% | 52 | 22.3% |
| Other | 19 | 8.15% | 10 | 4.29% | 9 | 3.86% |
| Not Reported | 6 | 2.58% | 3 | 1.29% | 3 | 1.29% |
| **Previous research experience** | | | | | | |
| Yes | 101 | 43.3% | 40 | 17.3% | 61 | 26.2% |
| No | 118 | 50.6% | 48 | 20.6% | 70 | 30.0% |
| Not Reported | 14 | 6.01% | 4 | 1.72% | 10 | 4.29% |
| **Proximity to medical appointments** | | | | | | |
| Less than 30 minutes | 58 | 24.9% | 24 | 10.3% | 34 | 14.6% |
| 30 minutes to 1 hour | 108 | 46.4% | 45 | 19.3% | 63 | 27.0% |
| More than 1 hour | 55 | 23.6% | 19 | 8.15% | 36 | 15.5% |
| Not Reported | 12 | 5.15% | 8 | 3,43% | 4 | 1.72% |
| **Medical Treatments Received** | **Body Area** | | | | | |
| | Arms | | Legs | | Spine/Trunk | |
| Bony surgery | 4 | 1.72% | 57 | 24.5% | 11 | 4.72% |
| Soft tissue surgery | 13 | 5.58% | 109 | 46.8% | 1 | 0.43% |
| Neural surgery | 1 | 0.43% | 6 | 2.58% | 18 | 7.73% |
| Botox or other injections | 39 | 16.7% | 72 | 30.9% | 6 | 2.58% |
| Non-injectable spasticity medication | 35 | 15.0% | 59 | 25.3% | 31 | 13.3% |
| Physical or occupational therapy (current) | 124 | 53.2% | 153 | 65.7% | 81 | 34.8% |
| Intensive therapy programs/camps (previous) | 42 | 18.0% | 39 | 16.7% | 20 | 8.58% |

*these demographics refer to the minor with cerebral palsy.

**Table 2. Summary metrics and statistical test results.**

| Variable | N | Summary Metrics Median (IQR) Mean (SD) | Statistical Test | p-values GMFCS Level | CP Type | Responder Type |
|---|---|---|---|---|---|---|
| **Personal Interest** | | | | | | |
| *Research importance:* Importance of CP research (0–100) | 229 | 99 (9) 93.8 (9.84) | | | | |
| *Value research participation:* Value of participation in CP research (0–100) | 221 | 97 (18) 88.2 (17.7) | | | | |
| *Compensation importance:* Importance of compensation for study participation (0–100) | 203 | 64 (29) 62.3 (25.7) | | | | |
| *Study types:* Study types most likely to contact a researcher to learn more about | 228 | *See Results: Personal Interests* | | | | |
| *Body functions:* Area of research focus most interested in participating or hearing more about | 233 | *See Fig 2A* | | | | |
| **Travel Needs & Preferences** | | | | | | |
| *Childcare:* Whether additional childcare is needed to participate | 221 | Yes: 75 No: 146 | Chi-square | p = 0.37 | p = 0.08 | **p<0.001** |
| *Time off work:* Whether time off work is needed to participate | 220 | Yes: 138 No: 82 | Chi-square | p = 0.18 | p = 0.39 | p = 0.31 |
| *Additional travel needs:* What other things need to be considered to travel to a research study | 174 | *See Fig 2B* | | | | |
| Time | | | Chi-square | p = 0.02 | **p = 0.003** | p = 0.91 |
| Breathing | | | Chi-square | **p = 0.002** | p = 0.43 | p = 0.70 |
| Transition | | | Chi-square | p = 0.02 | p = 0.25 | **p = 0.009** |
| Seizure | | | Chi-square | **p = 0.001** | p = 0.30 | **p = 0.002** |
| Feeding | | | Chi-square | **p<0.001** | **p<0.001** | **p<0.001** |
| Snacks | | | Chi-square | p = 0.48 | p = 0.13 | **p<0.001** |
| Medications | | | Chi-square | **p<0.001** | p = 0.03 | p = 0.88 |
| Toileting | | | Chi-square | **p<0.001** | **p<0.001** | **p<0.001** |
| Transportation | | | Chi-square | **p<0.001** | **p<0.001** | p = 0.22 |
| Other | | | Chi-square | p = 0.07 | p = 0.61 | p = 0.22 |
| *Transportation:* Preferred transportation method | 220 | *See Fig 2C* | | | | |
| Drive self | | | Chi-square | p = 0.01 | **p = 0.002** | **p<0.001** |
| Family member drives | | | Chi-square | p = 0.97 | p = 0.46 | p = 0.06 |
| Public transit | | | Chi-square | p = 0.02 | p = 0.09 | **p<0.001** |
| Ride service | | | Chi-square | p = 0.17 | p = 0.34 | **p<0.001** |
| Other | | | Chi-square | p = 0.03 | p = 0.06 | **p = 0.001** |
| *Travel time for indirect benefit study:* Maximum time willing to travel from home for study without the potential for direct benefit (0.5-more than 2 hrs) | 219 | 4 (3) 3.53 (1.40) | Chi-square | p = 0.26 | p = 0.44 | p = 0.06 |
| *Travel time for direct benefit study:* Maximum time willing to travel from home for study with the potential for direct benefit (0.5-more than 2 hrs) | 220 | 5 (2) 4.13 (1.19) | Chi-square | p = 0.37 | p = 0.15 | p = 0.12 |
| *Overnight trip:* Willingness to make overnight or extended trip for research study | 221 | Yes: 110 No: 19 Maybe: 92 | Chi-square | p = 0.98 | p = 0.69 | p = 0.75 |
| *Travel reimbursement for a local study:* Importance that cost of travel to local study is reimbursed (0–100) | 203 | 50 (57) 49.1 (32.8) | Kruskal-Wallis | p = 0.14 | p = 0.75 | **p = 0.007** |
| *Travel reimbursement for a distant study:* Importance that cost of travel to distant study is reimbursed (0–100) | 205 | 78 (32) 75.3 (24.3) | Kruskal-Wallis | p = 0.54 | p = 0.93 | p = 0.69 |
| **Study-specific Elements** | | | | | | |

*(Continued)*

**Table 2.** (Continued)

| Variable | N | Summary Metrics Median (IQR) Mean (SD) | Statistical Test | p-values | | |
|---|---|---|---|---|---|---|
| | | | | GMFCS Level | CP Type | Responder Type |
| *Locations:* Preferred study location | 222 | See *Fig 2D* | | | | |
| Current clinic | | | Chi-square | p = 0.68 | p = 0.62 | **p = 0.03** |
| New clinic | | | Chi-square | p = 0.29 | **p = 0.003** | p = 0.94 |
| Park | | | Chi-square | **p<0.001** | **p = 0.01** | **p = 0.02** |
| Lab | | | Chi-square | p = 0.58 | **p = 0.003** | p = 0.58 |
| School | | | Chi-square | p = 0.64 | p = 0.19 | **p<0.001** |
| Home | | | Chi-square | p = 0.12 | p = 0.95 | p = 0.18 |
| Other | | | Chi-square | p = 0.76 | p = 0.30 | p = 0.29 |
| *Time of year:* Times that would be considered for research participation | 127 | See *Fig 2E* | | | | |
| Weekends during school year | | | Chi-square | p = 0.11 | p = 0.06 | p = 0.68 |
| Weekdays during school year | | | Chi-square | p = 0.06 | p = 0.20 | **p = 0.05** |
| Summer break | | | Chi-square | p = 0.68 | p = 0.91 | **p<0.001** |
| Spring break | | | Chi-square | p = 0.46 | p = 0.61 | p = 0.11 |
| Winter break | | | Chi-square | p = 0.87 | p = 0.41 | **p = 0.002** |
| Non-attendance school days | | | Chi-square | p = 0.22 | p = 0.15 | **p = 0.002** |
| *Maximum time commitment:* Amount of time in one day that is reasonable to participate in a study (0.5–8 hrs) | 220 | 4 (3) <br> 3.83 (1.16) <br> See *Fig 2F* | Kruskal-Wallis | p = 0.11 | **p = 0.003** | **p = 0.002** |
| *Maximum study visits:* Maximum number of visits for one study (1–5 visits) | 219 | 5 (2) <br> 4.28 (1.10) | Kruskal-Wallis | p = 0.62 | p = 0.71 | p = 0.21 |
| *Longitudinal:* Willingness to participate in longitudinal study (Yes or No) | 220 | Yes: 210 <br> No: 10 | Chi-square | p = 0.19 | p = 0.94 | p = 0.19 |
| *Compensation amount:* Appropriate amount of compensation ($/hr) | 176 | 15 (10) <br> 16.7 (12.3) | Kruskal-Wallis | p = 0.71 | p = 0.68 | p = 0.94 |

excluded from analysis pertaining to that question. To summarize responses, descriptive analyses were first completed, with all percentages reported relative to the number of respondents for each question. We defined 50% as the threshold to describe the majority of survey respondents. Statistical analyses were only performed on travel and study-specific variables, as researchers can directly use this information to modify study methods during the developmental pipeline. Q-Q plots were created for quantitative variables to assess normality. Kruskal-Wallis tests, Chi-squared tests, or Kaplan-Meier survival analyses were performed on variables hypothesized to be dependent on three factors: Responder type (2 levels: adult, caregiver), CP type (4 levels: hemiplegia, diplegia, quadriplegia, other), and GMFCS level (5 levels: I, II, III, IV, V). A p-value < 0.05 was considered significant. Post hoc analyses were used to determine significant pairwise comparisons, where p-values were corrected for multiple comparisons using Bonferroni corrections. Further analyses were run to test specific hypotheses. For open-ended questions pertaining to personal interests, AH reviewed all responses, identified common themes, and categorized each response accordingly. Categorizations were reviewed and approved by KMZ and VG and summarized semi-quantitatively. For travel preferences, Wilcoxon signed-rank tests were run to determine differences between *travel time for indirect benefit* and *travel time for direct benefit* and between importance of *travel reimbursement for local study* and *travel reimbursement for distant study.*

## Results

### I. Demographics

In total, 255 individuals were consented and 233 (91.4% response rate) at least partially completed the survey. Respondent demographics are listed in Table 1. The survey population closely matches US census data in terms of sex, ethnicity, and race [14–16]. The majority of participants reported no previous research experience (53.9%), a proximity to medical appointments of 1 hour or less (75.1%), and previous or ongoing physical/occupational therapy treatment for their arms (53.2%) or legs (65.7%).

### II. Personal interests

All subsequent variables and associated p-values are listed in Table 2. Respondents reported high mean scores for *research importance* (93.8/100), *value research participation* (88.2/100), and *compensation importance* (62.3/100). Open-ended questions revealed that the biggest personal motivators for CP research were *personal benefit* (62.2% of respondents) and *helping others* (53.4%) (Fig 1A), while the biggest personal barrier was *schedule limitations* (48.9%) (Fig 1B). With regard to research interests, the most popular *study types* were physical or occupational therapy treatments (90.8%), activity monitoring (79.1%), imaging of muscle/bone (72.4%), survey or online (71.5%), robotic games (68.9%), imaging of the brain (68.4%), and new treatments (64.9%). The most popular *body functions* of interest were the legs/feet (79.9%), muscles (79.5%), movement/fitness (79.0%), brain/nerves (76.0%), arms/hands (62.9%), and pain (50.7%) (Fig 2).

### III. Travel needs

Most respondents needed *time off work* (62.7%) but did not need additional arrangements for *childcare* to engage in research (66.1%). There was a significant main effect of responder type on the latter, where caregivers of minors with CP needed *childcare* more than adults with CP. When leaving their homes, the majority of respondents had *additional travel needs* such as *time* (59.2%), *transportation items* such as a wheelchair or stroller (58.0%), and *snacks* (52.9%) (Fig 3A).

Specific *additional travel needs* varied significantly based on GMFCS level, CP type, and responder type. All significant pairwise comparisons are reported in the (S1 Table). There was a significant main effect of GMFCS level on *breathing items*, *seizure items*, *feeding items*, *medications*, *toileting items*, and *transportation items*. In summary, individuals classified as GMFCS level V reported needing these items more to comfortably travel. There was a significant main effect of CP type on *time*, *feeding items*, *toileting items*, and *transportation items*. In general, individuals affected by quadriplegia reported needing these items more to comfortably travel. Finally, there was a significant main effect of responder type on *transition*, *seizure items*, *feeding items*, *snacks*, and *toileting items*. Adults with CP had more concerns about *transition* to a new environment than caregivers of minors with CP. However, caregivers needed *seizure items*, *feeding items*, *snacks*, and *toileting items* for their children more than adults with CP did for themselves.

### IV. Travel preferences

The most common mode of *transportation* was by car, whether the individual drives (63.2%) or gets a ride from a family member (35.0%) (Fig 3B). There was a significant main effect of GMFCS level on *transportation* methods including *drive self*, *public transit*, and *other*, though there were no significant pairwise comparisons. There was a significant main effect of CP type

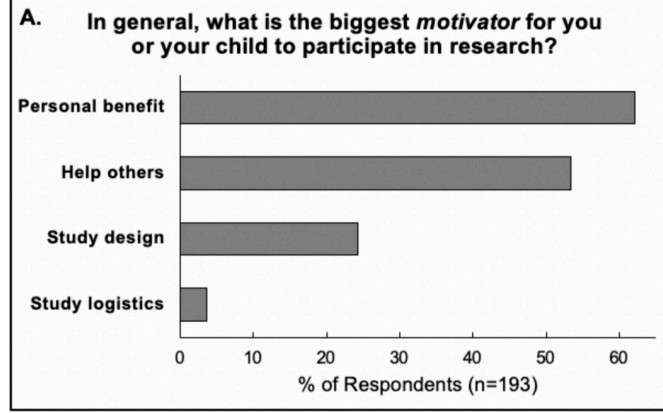

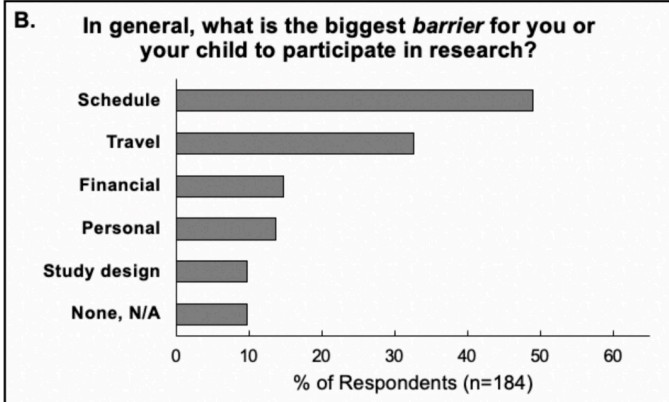

**Fig 1.** Percentage of respondents who indicated (A) motivators and (B) barriers for participating in research relating to the categories shown. Indented categories are subcategories of the parent category. (C) Representative quotes indicating goals for participating in research.

on *drive self*, where respondents (both caregivers and adults with CP) affected by hemiplegia preferred to drive themselves more than those affected by diplegia or quadriplegia. There was a significant main effect of responder type on *drive self*, *public transit*, *ride service*, and *other*. Caregivers of minors with CP preferred to drive themselves more than adults with CP, whereas adults preferred public transit, ride services, or other methods of transportation.

The mean response for *travel time for indirect benefit study* was 3.53 hours, which was significantly lower ($z = -6.857$, $p < 0.001$) than the mean response for *travel time for direct benefit study* at 4.13 hours. Approximately half (49.8%) of respondents were willing to make an *overnight trip* for research participation. There was a significant main effect of responder type on the importance of *travel reimbursement for a local study*, where adults with CP thought reimbursement was more important than caregivers of minors with CP. The overall mean score for

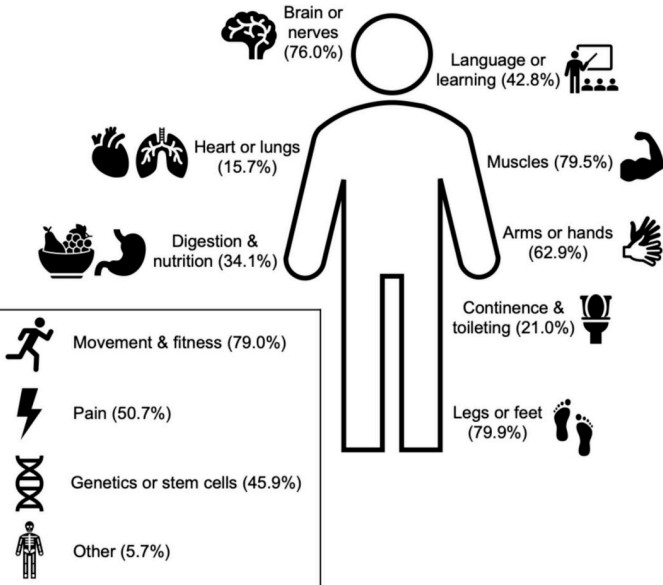

**Fig 2.** *Body functions* **of research interest to respondents.** Each of these options was offered as a checkbox for respondents to indicate if they would be interested in participating in a study that focused on these body regions/functions. Percentages are out of n = 233 respondents.

this variable (49.1/100) was significantly lower (z = -9.901, p < 0.001) than the mean score for the importance of *travel reimbursement for a distant study* (75.3/100).

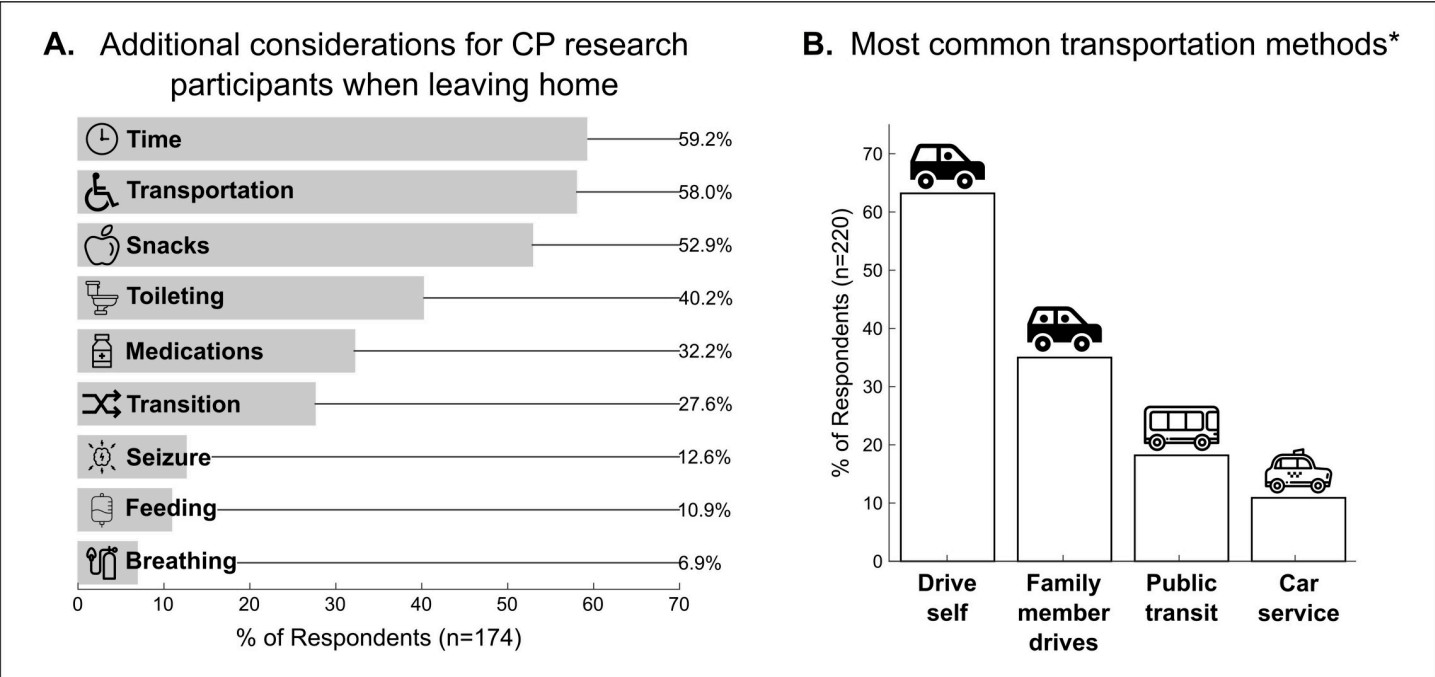

**Fig 3.** (A) Summary of *additional travel needs* required to participate in CP research. The most prevalent categories were *time*, *transportation*, and *snacks*. (B) Summary of the most common *transportation* methods. Driving was the most cited mode of transportation. *6.8% of respondents selected other transportation modes.

**Study-specific elements.** A *current clinic* (88.7%), *home* (84.7%), *lab* (73.0%), and *new clinic* (59.9%) were the most preferred *locations* for research participation (Fig 4A). There was a significant main effect of GMFCS level on *park*, where individuals who are GMFCS level I were more likely to select this location than all other levels. There was a significant main effect of CP type on *new clinic* and *lab*, where individuals affected by hemiplegia were more likely to select these locations over those affected by quadriplegia. There was a significant main effect of responder type on *current clinic*, *park*, and *school*, where caregivers of minors with CP were more likely to select these locations over adults with CP.

The majority of respondents were flexible to participate in research at any *time of year*, except for parents on *weekdays during the school year* (Fig 4C). There was a significant main effect of responder type on weekdays, where adults reported more availability compared to caregivers. There was also a significant main effect of responder type for *summer break*, *winter break*, and *other school holidays*, where caregivers were more willing to engage in research during these *times of year* than adults. Respondents indicated a mean *compensation amount* of $16.69/hour for participation in research. The average *maximum time commitment* was 3.83 hours/day and the average *maximum study visits* was 4.28 visits. Survival analyses on *maximum time commitment* yielded significant main effects of CP type (log rank $\chi^2(1) = 11.9$, p = 0.001), with no significant pairwise comparisons, and responder type (log rank $\chi^2(3) = 13.4$, p = 0.004). Most notably, caregiver interest dropped from 54.1% to 24.1% at a *maximum time commitment* greater than 4 hours (Fig 4B). Finally, the vast majority of respondents were willing to participate in a *longitudinal* study (95.5%).

## Discussion

The purpose of this study was to determine the motivators and barriers involved in the decision to participate in CP research studies. We administered a survey to gain more insight on stakeholder perspectives, including their personal interest in research, travel needs and preferences for study participation, and study-specific elements. These results can be extrapolated into recommendations for future CP research studies to maximize participant recruitment and expedite new knowledge about CP.

Our study is one of the first to identify the personal and practical factors that influence research participation. Respondents overwhelmingly supported research and valued their own participation in research, in a wide range of topics (Fig 2A). The research areas of interest identified by our survey respondents are largely consistent with previous reports of CP research priorities [8, 17]. Survey responses indicated that most individuals were motivated to participate by the potential for personal benefit and helping others. Where relevant, these two elements should be highlighted in recruitment materials and results should be disseminated to participants in a format that is best suited to their learning preferences (e.g. a copy of a manuscript or poster, a one-page summary, or a short video). Schedule limitations were the most prevalent barriers to research participation, especially as caregivers of minors with CP were likely to need additional childcare arrangements. Travel limitations were also a highly cited barrier to research participation. To minimize these barriers, researchers should offer flexible study times, particularly during the weekends and summer break, and/or utilize local study locations close to the home or clinics where participants are receiving care.

When scheduling participants with CP for a research study located outside of their home, travel needs for participants should be considered. To accommodate the additional time required for participants and/or their families to reach the study location, researchers should be flexible with appointment times. Indeed, utilizing flexible study protocols has previously been identified as a recommendation to improve recruitment in CP research [6]. Researchers should

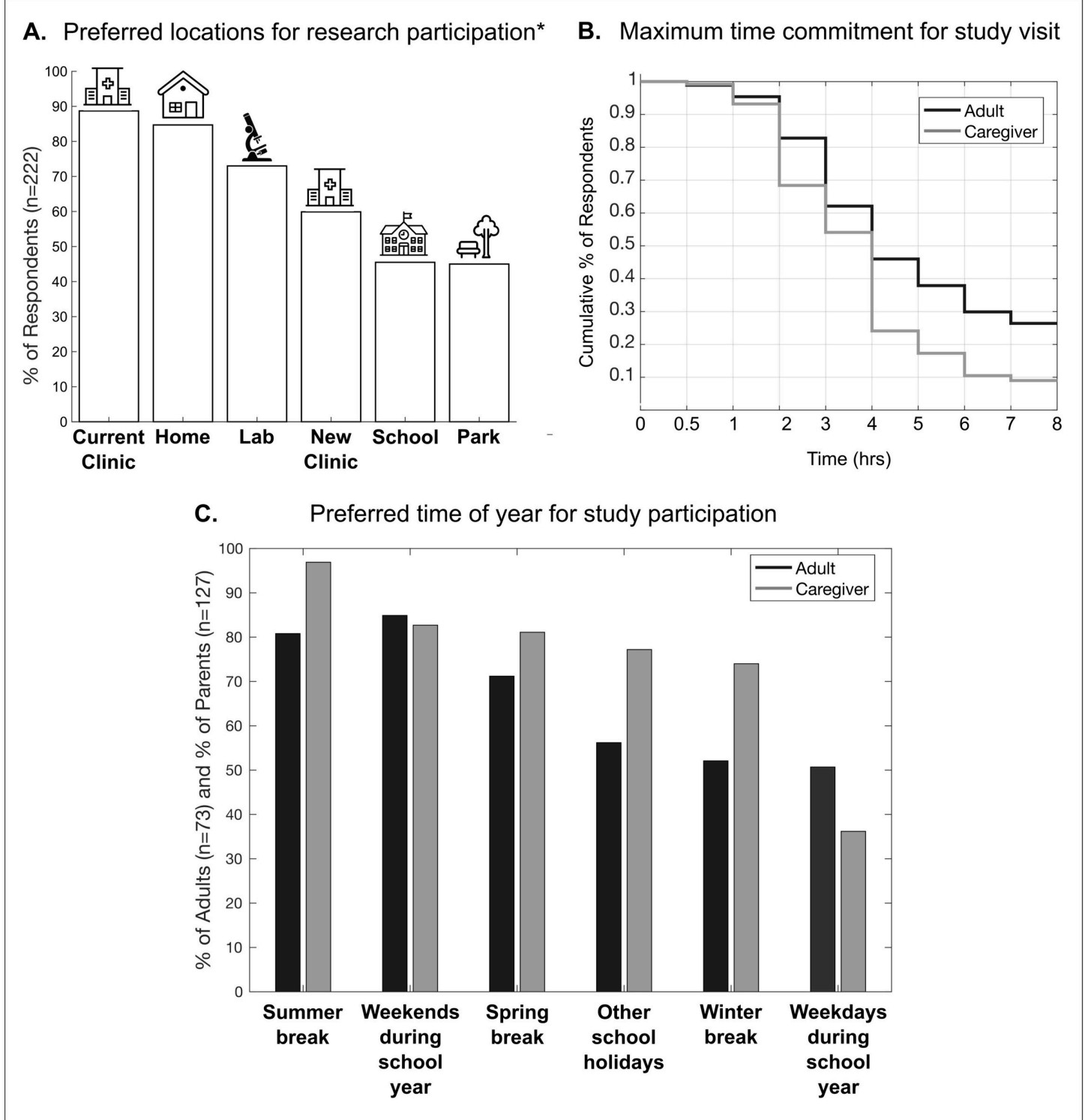

**Fig 4.** (A) Summary of preferred *locations* for CP studies. *5.4% of respondents chose other locations. (B) Survival analysis for *maximum time commitment* by responder type. Less than 30% of caregivers and adults remain at 4 and 7 hours, respectively. (C) Summary of preferred *time of year* for participation by responder type.

also consider having snacks available for participants, particularly for minors with CP. Unsurprisingly, participants classified as GMFCS level V or with a diagnosis of quadriplegia reported requiring more items in order to comfortably travel. Researchers working with these inclusion criteria might consider the home as a study location to mitigate the barrier of travel burden.

Respondents clearly noted the importance of compensation, as providing compensation to participants is consistent with appreciation of their time. In a previous study where caregivers of minors with CP were consulted on the design of randomized control trials, there was a strong preference for coverage of all treatment costs [18]. Caregivers noted that they may not be able to cover the costs themselves, the participants would be offering their time, and the study benefits were unknown [18]. From our survey responses, a minimum compensation of $15/hour and a maximum time commitment of 4 hours/day were interpreted to be respectful of the time and commitment to research participation.

In addition to financial incentives for participation, compensation for other expenses associated with travel should be considered. If a study session requires a longer duration, additional compensation can include paying for a meal. Because most respondents were willing to travel long distances for studies with and without potential for direct benefit, researchers should also consider offering travel reimbursement. This is especially important because travel limitations were a highly cited barrier to research participation. As the most preferred mode of transportation involves driving, suggestions for reimbursement include gas and parking. Flexibility around transport mode could also include fare coverage for adult participants who prefer public transit or ride services. For participants willing to make an overnight trip for a research study, researchers should consider compensating for lodging and overnight parking.

Our survey-based recommendations are centered around maximizing stakeholder participation in CP research studies. One limitation of our sample was that survey respondents were self-selected and may be biased towards research participation. Their responses may inflate measures of research importance and resource allocation (e.g. time and money), while underestimating obstacles to research participation. However, these individuals may also be more likely to respond to participant requests and therefore would be more representative of future study samples. Another limitation is that we allowed the terms "potential for direct benefit" and "indirect benefit" to be interpreted by the respondents. This does not address considerations such as the variability of perception of direct benefit [19] and therapeutic misconceptions [20]. Survey responses were self-reported at one time point. Future research should determine whether attitudes towards research shift over time or are dependent on the depth of previous research experience. Finally, researchers should be informed about and supported in the engagement of the community in research. This can span a spectrum of involvement, such as one-time consultations to provide feedback on study-specific elements, formation of Community Advisory Boards, or the inclusion of community stakeholders as project investigators. Our study including stakeholders during the design phase but would have required additional funding to adequately reimburse time and efforts for larger scale engagement. As a research team, we continue to look for ways to include family stakeholders in the research process as equal partners.

We assessed the motivators and barriers to research participation from the perspectives of caregivers of minors with CP and adults with CP. By identifying these stakeholder attitudes and utilizing the information to design study protocols, individuals with CP and their families become true partners in the research that aims to benefit people like them. Researchers can best accommodate the needs of participants with CP by opting for flexible study locations, scheduling, and compensation options. We recognize that this will not be feasible for all studies but encourage researchers to consider even the smallest gestures to reduce the burden of participation.

## Supporting information

**S1 Table. Significant pairwise comparisons after Bonferroni correction.** *Abbreviations:
GMFCS = Gross Motor Function Classification System, di = diplegia, hemi = hemiplegia,
quad = quadriplegia.
(DOCX)

**S1 File.**
(DOCX)

## Acknowledgments

The authors would like to thank our research colleagues Divya Joshi, Nayo Hill, and Heidi
Roth for their help in the design of survey questions, recruitment of participants, and discussions of interpretation of responses. In addition, we acknowledge the contributions of community stakeholders for their feedback on survey design.

## Author Contributions

**Conceptualization:** Theresa Sukal-Moulton.

**Data curation:** Theresa Sukal-Moulton.

**Formal analysis:** Kristina M. Zvolanek, Vatsala Goyal, Alexandra Hruby.

**Investigation:** Kristina M. Zvolanek, Vatsala Goyal, Alexandra Hruby, Carson Ingo, Theresa
Sukal-Moulton.

**Methodology:** Kristina M. Zvolanek, Vatsala Goyal, Alexandra Hruby, Theresa Sukal-
Moulton.

**Project administration:** Theresa Sukal-Moulton.

**Supervision:** Carson Ingo, Theresa Sukal-Moulton.

**Visualization:** Kristina M. Zvolanek, Vatsala Goyal, Alexandra Hruby.

**Writing – original draft:** Kristina M. Zvolanek, Vatsala Goyal.

**Writing – review & editing:** Kristina M. Zvolanek, Vatsala Goyal, Alexandra Hruby, Carson
Ingo, Theresa Sukal-Moulton.

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
