## [Decision Letter · Decision Letter 0]

7 Sep 2021

PONE-D-21-23605Motivators and barriers to research participation for individuals with cerebral palsy and their familiesPLOS ONE

Dear Dr. Sukal-Moulton,

Thank you for submitting your manuscript to PLOS ONE. After careful consideration, we feel that it has merit but does not fully meet PLOS ONE’s publication criteria as it currently stands. Therefore, we invite you to submit a revised version of the manuscript that addresses the points raised during the review process.

Please submit your revised manuscript by Oct 22 2021 11:59PM. Please include the following items when submitting your revised manuscript:A 'Response to Reviewers' letter that responds to each point raised by the academic editor and reviewer(s). You should upload this letter as a separate file labeled 'Response to Reviewers'.A marked-up copy of your manuscript that highlights changes made to the original version. You should upload this as a separate file labeled 'Revised Manuscript with Track Changes'.An unmarked version of your revised paper without tracked changes. You should upload this as a separate file labeled 'Manuscript'.

We look forward to receiving your revised manuscript.

Kind regards,

Prof. Ritesh G. Menezes, M.B.B.S., M.D., Diplomate N.B.

Academic Editor

PLOS ONE

Reviewers' comments:

Reviewer's Responses to Questions

**Comments to the Author**

1. Is the manuscript technically sound, and do the data support the conclusions?

Reviewer #1: Yes

Reviewer #2: Yes

2. Has the statistical analysis been performed appropriately and rigorously? 

Reviewer #1: Yes

Reviewer #2: Yes

3. Have the authors made all data underlying the findings in their manuscript fully available?

Reviewer #1: Yes

Reviewer #2: Yes

4. Is the manuscript presented in an intelligible fashion and written in standard English?

Reviewer #1: Yes

Reviewer #2: Yes

5. Review Comments to the Author

Reviewer #1: The manuscript is very good, the essence and concept are very clear and precise. A topic of this reserarch has importance in the motivations and increase number of respondents in the future research. It will be useful for future researchers in these fields.

Reviewer #2: Thank you for the opportunity to review this interesting article. It is a worthwhile contribution to the literature, to understand the facilitators and barriers to people with CP and their families participating in research.

Excellent introduction, sets up the study well.

Methods: Considering the topic, even though opinions were sought, it seems a shame there were no people with CP or family members involved as partners in this research? I see now that three family members seem to be acknowledged. Why are they not authors if they were involved in design and recruitment? Did you discuss interpretation of responses with them?

How were the categories and questions decided upon? Were families involved in this or did they just review for readability?

How was a scale of 1-100 decided upon?

The authors don't seem to have used a checklist such as CHERRIES https://www.ncbi.nlm.nih.gov/pmc/articles/PMC1550605/

Results: A good number of responses (although response rate not reported). It would be helpful if % were written in Table 1, not just numbers. Motor type refers to spasticity/dyskinesis etc, but here it refers to topography.

Table 2 is difficult to interpret, especially what the three right hand columns are referring to. This needs further explanation. Is the main finding that they differ or don't differ? We are unable to tell from the table what the difference is (i.e. with GMFCS which GMFCS levels are associated with the variable of interest).

Discussion: There is limited literature supporting or critiquing the findings throughout the discussion. However, there are some very helpful findings from this survey, congratulations to the investigators.

6. PLOS authors have the option to publish the peer review history of their article (what does this mean?). If published, this will include your full peer review and any attached files.

Reviewer #1: No

Reviewer #2: No

---

## [Author Response · Author response to Decision Letter 0]

14 Oct 2021

Reviewer(s)' Comments to Authors and Responses 

(1) The manuscript is very good, the essence and concept are very clear and precise. A topic of this research has importance in the motivations and increases the number of respondents in the future research. It will be useful for future researchers in these fields.

Thank you for your comments and for taking the time to review the manuscript.

(2) Thank you for the opportunity to review this interesting article. It is a worthwhile contribution to the literature, to understand the facilitators and barriers to people with CP and their families participating in research.

Thank you for this encouraging comment.

(3) Excellent introduction, sets up the study well.

Thank you.

(4) Methods: Considering the topic, even though opinions were sought, it seems a shame there were no people with CP or family members involved as partners in this research? I see now that three family members seem to be acknowledged. Why are they not authors if they were involved in design and recruitment? Did you discuss interpretation of responses with them?

We appreciate this thoughtful comment. As we’ve now clarified in the acknowledgements section, the three individuals listed are research colleagues of the authors, and we have rectified the accidental omission those affected by CP in this section. Paragraph 2 of the methods section explains that we sought feedback from six individuals affected by CP on the clarity of survey questions, the completeness of content, and the importance of the survey goals. While these individuals were incredibly generous with their time, there were timing constraints associated with the pandemic that made a more substantial commitment difficult. Furthermore, we did not have funding to appropriately compensate them for larger contributions to the work. In relation to those affected by CP that participated in the survey, we plan to disseminate the results to all individuals who consented and selected the optional element to be informed.

(5) How were the categories and questions decided upon? Were families involved in this or did they just review for readability?

The original concept for this study was borne from organic discussions among our research team, so the objectives of the study were mostly established before engaging with stakeholders. However, we did make it clear that we were open to feedback at all levels and the stakeholders who responded focused on the way that questions were asked and how the survey was functioning. We clarified this in paragraph 2 of the methods section.

(6) How was a scale of 1-100 decided upon?

We used the REDCap digital version of the visual analog scale (VAS) for three personal interests variables (research importance, value research participation, compensation importance) to easily quantify subjective opinions, similar to the VAS used in pain literature. We’ve updated this in paragraph 4 of the methods section.

(7) The authors don't seem to have used a checklist such as CHERRIES https://www.ncbi.nlm.nih.gov/pmc/articles/PMC1550605/

Thank you for bringing this to our attention. We completed the CHERRIES checklist and uploaded it as a separate document.

(8) Results: A good number of responses (although response rate not reported). It would be helpful if % were written in Table 1, not just numbers. Motor type refers to spasticity/dyskinesis etc., but here it refers to topography.

Overall response rate (91.4%) has been clarified in the first paragraph of the Results section. Table 1 has been modified to include the percentage of respondents in addition to the numbers for each demographic. Per the reviewer’s suggestion, “motor type” was changed to “motor topography” to reflect the information more appropriately in this category of Table 1.

(9) Table 2 is difficult to interpret, especially what the three right hand columns are referring to. This needs further explanation. Is the main finding that they differ or don't differ? We are unable to tell from the table what the difference is (i.e. with GMFCS which GMFCS levels are associated with the variable of interest).

Thank you for your remarks regarding the clarity of Table 2. We have removed the test statistics and degrees of freedom from each column to make the table more digestible, and we added a column to specify the statistical test used for each variable. Significant results in the three right-hand columns indicate differences between the particular respondent grouping (e.g. GMFCS levels, CP types) and the variable of interest. Please note that supplemental Table S1, which is referenced in the text (second paragraph of the ‘Travel Needs’ Results section), lists the significant pairwise comparisons. This table can be referenced for specific differences in GMFCS levels, etc. for each variable.

(10) Discussion: There is limited literature supporting or critiquing the findings throughout the discussion. However, there are some very helpful findings from this survey, congratulations to the investigators.

The reviewer raises a valid point; we are limited by the sparsity of existing literature on this topic. We have done our best to identify and discuss relevant previous work. This is all the more reason to do similar studies to gain a broader understanding of the needs and priorities of the cerebral palsy community.

---

## [Decision Letter · Decision Letter 1]

22 Nov 2021

PONE-D-21-23605R1Motivators and barriers to research participation for individuals with cerebral palsy and their familiesPLOS ONE

Dear Dr. Sukal-Moulton,

Thank you for submitting your manuscript to PLOS ONE. After careful consideration, we feel that it has merit but does not fully meet PLOS ONE’s publication criteria as it currently stands. Therefore, we invite you to submit a revised version of the manuscript that addresses the points raised during the review process.

Please submit your revised manuscript by 30-November-2021. Please include the following items when submitting your revised manuscript:A 'Response to Reviewers' letter that responds to each point raised by the academic editor and reviewer(s). You should upload this letter as a separate file labeled 'Response to Reviewers'.A marked-up copy of your manuscript that highlights changes made to the original version. You should upload this as a separate file labeled 'Revised Manuscript with Track Changes'.An unmarked version of your revised paper without tracked changes. You should upload this as a separate file labeled 'Manuscript'.

We look forward to receiving your revised manuscript.

Kind regards,

Prof. Ritesh G. Menezes, M.B.B.S., M.D., Diplomate N.B.

Academic Editor

PLOS ONE

Journal Requirements:

Reviewers' comments:

Reviewer's Responses to Questions

**Comments to the Author**

1. If the authors have adequately addressed your comments raised in a previous round of review and you feel that this manuscript is now acceptable for publication, you may indicate that here to bypass the “Comments to the Author” section, enter your conflict of interest statement in the “Confidential to Editor” section, and submit your "Accept" recommendation.

Reviewer #2: All comments have been addressed

Reviewer #3: (No Response)

Reviewer #4: (No Response)

2. Is the manuscript technically sound, and do the data support the conclusions?

Reviewer #2: Yes

Reviewer #3: Yes

Reviewer #4: Yes

3. Has the statistical analysis been performed appropriately and rigorously? 

Reviewer #2: Yes

Reviewer #3: I Don't Know

Reviewer #4: I Don't Know

4. Have the authors made all data underlying the findings in their manuscript fully available?

Reviewer #2: Yes

Reviewer #3: Yes

Reviewer #4: Yes

5. Is the manuscript presented in an intelligible fashion and written in standard English?

Reviewer #2: Yes

Reviewer #3: Yes

Reviewer #4: Yes

6. Review Comments to the Author

Reviewer #2: Thank you for your responses to the reviewers. I have nothing further to add, and look forward to seeing the paper in the literature. We all have a lot to improve on in this space.

Reviewer #3: page 8, line 332-333 can you please add more clarity to this sentence: "In addition to direct payment, compensation for other costs associated with travel should be considered." There is an ethical difference between giving someone an incentive to sign up for a research study (e.g., $50 gift card), and reimbursing their expenses (e.g., travel, parking). What exactly do you mean by "direct payment"? In your paper, please be clear on use of words that are meant to describe a recruitment incentive and expense compensation.

Reviewer #4: First, I would like to thank the authors for their patience during my review of this manuscript. I personally thank the authors for conducting this thoughtful survey - despite the pandemic setbacks.

I recommend this encouraging paper for acceptance to PLoS ONE. Recognition of the obstacles that caregivers and individuals with CP face when being included in research is a necessary addition to the literature and empowerment of the disability community.

A few comments and suggestions below:

1. In general, there is a narrative of "indirect benefit" vs "direct benefit" in the Travel Needs and Preferences category under Part III of Methods section. From my understanding, research does not directly benefit those who participate (that would be more of a QI initiative) but rather in some instances there may be the prospect of benefit. Authors can read more here: https://www.ncbi.nlm.nih.gov/pmc/articles/PMC2945615/

Friedman et al argue about “direct” and “indirect” benefits in the context of research on vulnerable populations and children and try to outline what counts as a potential direct benefit, if any.

2. “Research” is referred to broadly. Maybe in the future it would be useful to understand if the barriers to research are associated with the type of research involvement (e.g., research comparing routine standard of care interventions vs research on pre-approval CP therapeutics).

3. Page 14, line 255: Mean response time is reported for “travel time for indirect benefit study” vs “travel time for direct benefit study”. Again, it’s still not clear if this is the participant's perception of the benefit that they may derive from their participation in the study? Research aims to benefit future patients and often participation is altruistic, where research outcomes can benefit the scientific understanding of CP for future treatment and intervention. There is uncertainty around the interventions/therapies being studied. Your findings regarding the barrier “travel time” in light of “direct” vs “indirect” benefits could possibly help ethics review committees make prospective decisions on whether the research in question offers an appropriate risk-benefit profile, as certain prospects of benefit can be seen as decreasing impact of barriers and maybe even the risk involved to those participating. More studies like this survey are recommended.

I think a brief explanation on what the authors mean by "indirect benefit study" and "direct benefit study" would be meaningful. Maybe an inherent barrier is education regarding the purpose of research, since "indirect benefit" and "direct benefit" is seen impacting barriers like travel time. Mention of therapeutic misconception (inflating likelihood of benefit from research participation by seeing research as routine care) would be useful. Read here: https://www.ncbi.nlm.nih.gov/pmc/articles/PMC3690536/

5. Page 18, line 335: Again, is this their perception of benefit?

6. Page 18, line 352: This is an excellent point among many. Valuable and well-designed research should be informed by stakeholder and patient perspectives.

7. Limitations are well addressed. Great solutions proposed on CP stakeholder engagement.

8. Page 19, line 363: “…true partners in research that aims to benefit *people like* them."

Congratulations to the authors on this important work!

7. PLOS authors have the option to publish the peer review history of their article (what does this mean?). If published, this will include your full peer review and any attached files.

Reviewer #2: **Yes: **Sarah McIntyre

Reviewer #3: No

Reviewer #4: No

---

## [Author Response · Author response to Decision Letter 1]

14 Dec 2021

Reviewer(s)' Comments to Authors and Responses 

(1) Thank you for your responses to the reviewers. I have nothing further to add and look forward to seeing the paper in the literature. We all have a lot to improve on in this space.

We appreciate your comments and for taking the time to review the manuscript.

(2) Page 8, line 332-333 can you please add more clarity to this sentence: "In addition to direct payment, compensation for other costs associated with travel should be considered." There is an ethical difference between giving someone an incentive to sign up for a research study (e.g., $50 gift card), and reimbursing their expenses (e.g., travel, parking). What exactly do you mean by "direct payment"? In your paper, please be clear on the use of words that are meant to describe a recruitment incentive and expense compensation.

Thank you for your comments. We intended “direct payment” to be defined as the gift card or check that participants receive as incentive for study participation, but understand the concern that you’re sharing. Therefore, we have edited the sentence per your suggestion to: “In addition to financial incentives for participation, compensation for other expenses associated with travel should be considered” (pg. 18, lines 340-341 in the manuscript with tracked changes).

(3) First, I would like to thank the authors for their patience during my review of this manuscript. I personally thank the authors for conducting this thoughtful survey - despite the pandemic setbacks. I recommend this encouraging paper for acceptance to PLOS ONE. Recognition of the obstacles that caregivers and individuals with CP face when being included in research is a necessary addition to the literature and empowerment of the disability community.

Thank you for your kind comments and for taking the time to review the manuscript.

(4.1) In general, there is a narrative of "indirect benefit" vs "direct benefit" in the Travel Needs and Preferences category under Part III of Methods section. From my understanding, research does not directly benefit those who participate (that would be more of a QI initiative) but rather in some instances there may be the prospect of benefit. Authors can read more here: https://www.ncbi.nlm.nih.gov/pmc/articles/PMC2945615/. Friedman et al argue about “direct” and “indirect” benefits in the context of research on vulnerable populations and children and try to outline what counts as a potential direct benefit, if any.

Thank you for this insightful comment. While we agree with the definitions outlined by Friedman et al., our approach differed in the survey design. We have added definitions of the “potential for a direct benefit study” vs. an “indirect benefit study” using the language that was in our survey (pg. 6, lines 148-152 and Table 2 in the manuscript with tracked changes). Our intention when asking these questions was to differentiate between studies such as clinical trials (potential for direct benefit) and basic science investigations (indirect benefit). We agree that we did not control for participants’ understanding and interpretation of the likelihood for direct benefit, which is the rationale for using the same definition for readers. This limitation has been addressed in the discussion section (pg. 18, lines 357-360).

(4.2) Page 14, line 255: Mean response time is reported for “travel time for indirect benefit study” vs “travel time for direct benefit study”. Again, it’s still not clear if this is the participant's perception of the benefit that they may derive from their participation in the study? Research aims to benefit future patients and often participation is altruistic, where research outcomes can benefit the scientific understanding of CP for future treatment and intervention. There is uncertainty around the interventions/therapies being studied. Your findings regarding the barrier “travel time” in light of “direct” vs “indirect” benefits could possibly help ethics review committees make prospective decisions on whether the research in question offers an appropriate risk-benefit profile, as certain prospects of benefit can be seen as decreasing impact of barriers and maybe even the risk involved to those participating. More studies like this survey are recommended.

We have provided the definitions of “potential for direct benefit” vs. “indirect benefit” according to the language in the survey (pg. 6, lines 148-152 and Table 2). Our question about travel time was related to convenience rather than risk and is therefore likely less relevant to an ethics committee determination. We agree that how benefits are perceived by potential participants is important and should be studied in the future.

(4.3) I think a brief explanation on what the authors mean by "indirect benefit study" and "direct benefit study" would be meaningful. Maybe an inherent barrier is education regarding the purpose of research, since "indirect benefit" and "direct benefit" is seen impacting barriers like travel time. Mention of therapeutic misconception (inflating likelihood of benefit from research participation by seeing research as routine care) would be useful. Read here: https://www.ncbi.nlm.nih.gov/pmc/articles/PMC3690536/

As mentioned above, we defined these terms very broadly for participants. We have edited the limitations section of the discussion to acknowledge that we did not address considerations such as therapeutic misconception (pg. 18, lines 357-360). We agree that this is an important topic for investigation in future studies.

(4.4) Page 18, line 335: Again, is this their perception of benefit?

Yes, this is the participants’ perception of benefit. We have addressed this by clarifying the phrase in the relevant sentence: “...studies with and without potential for direct benefit” (pg. 18, line 343).

(5) “Research” is referred to broadly. Maybe in the future it would be useful to understand if the barriers to research are associated with the type of research involvement (e.g., research comparing routine standard of care interventions vs research on pre-approval CP therapeutics).

Thank you for this comment. We agree that this would be an interesting direction for future exploration.

(6) Page 18, line 352: This is an excellent point among many. Valuable and well-designed research should be informed by stakeholder and patient perspectives.

Thank you for this kind comment.

(7) Limitations are well addressed. Great solutions proposed on CP stakeholder engagement.

Thank you, we’re glad to see a shift towards this engaged approach occurring in research and hope to accelerate it. 

(8) Page 19, line 363: “…true partners in research that aims to benefit *people like* them."

This sentence has been edited as proposed (pg. 19, lines 378-

---

## [Decision Letter · Decision Letter 2]

17 Dec 2021

Motivators and barriers to research participation for individuals with cerebral palsy and their families

PONE-D-21-23605R2

Dear Dr. Sukal-Moulton,

We’re pleased to inform you that your manuscript has been judged scientifically suitable for publication and will be formally accepted for publication once it meets all outstanding technical requirements.

Kind regards,

Prof. Ritesh G. Menezes, M.B.B.S., M.D., Diplomate N.B.

Academic Editor

PLOS ONE

Reviewers' comments:

Reviewer's Responses to Questions

**Comments to the Author**

1. If the authors have adequately addressed your comments raised in a previous round of review and you feel that this manuscript is now acceptable for publication, you may indicate that here to bypass the “Comments to the Author” section, enter your conflict of interest statement in the “Confidential to Editor” section, and submit your "Accept" recommendation.

Reviewer #3: All comments have been addressed

2. Is the manuscript technically sound, and do the data support the conclusions?

Reviewer #3: (No Response)

3. Has the statistical analysis been performed appropriately and rigorously? 

Reviewer #3: (No Response)

4. Have the authors made all data underlying the findings in their manuscript fully available?

Reviewer #3: (No Response)

5. Is the manuscript presented in an intelligible fashion and written in standard English?

Reviewer #3: (No Response)

6. Review Comments to the Author

Reviewer #3: (No Response)

7. PLOS authors have the option to publish the peer review history of their article (what does this mean?). If published, this will include your full peer review and any attached files.

Reviewer #3: No

---

## [Editor Report · Acceptance letter]

13 Jan 2022

PONE-D-21-23605R2 

Motivators and barriers to research participation for individuals with cerebral palsy and their families 

Dear Dr. Sukal-Moulton:

I'm pleased to inform you that your manuscript has been deemed suitable for publication in PLOS ONE. Congratulations! Your manuscript is now with our production department. 

Kind regards, 

on behalf of

Prof. Dr. Ritesh G. Menezes 

Academic Editor

PLOS ONE